# A Case of *DNAJC12*-Deficient Hyperphenylalaninemia Detected on Newborn Screening: Clinical Outcomes from Early Detection

**DOI:** 10.3390/ijns10010007

**Published:** 2024-01-17

**Authors:** Colleen Donnelly, Lissette Estrella, Ilona Ginevic, Jaya Ganesh

**Affiliations:** Department of Genetics and Genomic Sciences, Icahn School of Medicine at Mount Sinai, New York, NY 10029, USA; lissette.estrella@mssm.edu (L.E.);

**Keywords:** hyperphenylalaninemia, DNAJC12, genetics, newborn screening, neurotransmitter deficiency, developmental delay, intellectual disability

## Abstract

*DNAJC12*-deficient hyperphenylalaninemia is a recently described inborn error of metabolism associated with hyperphenylalaninemia, neurotransmitter deficiency, and developmental delay caused by biallelic pathogenic variants of the *DNAJC12* gene. The loss of the *DNAJC12*-encoded chaperone results in the destabilization of the biopterin-dependent aromatic amino acid hydroxylases, resulting in deficiencies in dopamine, norepinephrine, and serotonin. We present the case of a patient who screened positive for hyperphenylalaninemia on newborn screening and was discovered to be homozygous for a likely pathogenic variant of *DNAJC12*. Here, we review the management of *DNAJC12*-related hyperphenylalaninemia and compare our patient to other reported cases in the literature to investigate how early detection and management may impact clinical outcomes.

## 1. Introduction

*DNAJC12*-deficient hyperphenylalaninemia is a newly described, autosomal recessively inherited neurotransmitter synthesis disorder, and biallelic pathogenic variants of *DNAJC12* have been identified in all patients [1,2]. This condition is characterized by deficiencies in multiple neurotransmitters, including dopamine, norepinephrine, and serotonin [1]. When untreated, this neurotransmitter deficiency results in symptoms including intellectual disability, developmental delay, autism spectrum disorder, and dystonia [1,2]. The protein encoded from the *DNAJC12* gene functions as a chaperone that participates in the proper folding of the biopterin-dependent aromatic amino acid hydroxylases, phenylalanine hydroxylase, tyrosine hydroxylase, and tryptophan hydroxylase, which are involved in the metabolism of phenylalanine, as well as the synthesis of a number of the aforementioned neurotransmitters [1,2]. In addition to elevated phenylalanine levels, low levels of biogenic amines, including homovanillic acid (HVA), 5-hydroxyindolacetic acid (5-HIAA), and 3 o-methyldopa in the cerebrospinal fluid (CSF), are characteristic biochemical features of this condition [1,2]. Here, we describe a patient with global developmental delay, hypotonia, and dysmorphic features who was initially brought to clinical attention after screening positive via newborn screening for hyperphenylalaninemia and ultimately diagnosed with *DNAJC12*-deficient hyperphenylalaninemia. We also review the literature for other reported cases of this condition to determine how earlier detection through newborn screening may have impacted this patient’s clinical outcome. 

## 2. Case Report

A newborn Hispanic female first came to the attention of our program following a positive newborn screen (NBS) for hyperphenylalaninemia. The first abnormal sample indicated a phenylalanine (Phe) level of 159.91 µM (reference < 151 µM) and a Phe/Tyrosine (Tyr) ratio of 7.42 (reference < 2.5). The subsequent sample indicated a Phe level of 259.7 µM and a Phe/Tyr ratio of 5.19. The patient was a fraternal twin birth, born premature at 25w2d gestation via C-section. The neonatal course was prolonged and complicated, and it included mechanical ventilation, surgical closure of a patent ductus arteriosus, and hyperalimentation, leading to a 96-day NICU admission at an outside hospital. Confirmatory biochemical testing confirmed mild hyperphenylalaninemia, indicating a Phe level of 163.2 µM (2.7 mg/dL). Phenylalanine and tyrosine levels remained in the therapeutic range (2–6 mg/dL) despite hyperalimentation, and subsequently, an unrestricted diet with formula was recommended. 

Genetic testing using a commercially available laboratory was conducted during the patient’s NICU admission and included the sequencing and deletion/duplication analysis of six genes related to hyperphenylalaninemia: *PAH*, *GCH1*, *PCBD1*, *QDPR*, *PTS*, and *SPR*. This multigene panel returned negative. At the time, this was the most comprehensive commercially available multigene panel to check for a genetic etiology for hyperphenylalaninemia. A high-resolution chromosomal microarray was also conducted, which returned results consistent with a normal female. Cofactor screening to assess for defects in the tetrahydrobiopterin pathway returned negative, and it appeared to suggest phenylalanine hydroxylase (PAH) enzyme deficiency. Additional testing was limited as compliance with onsite clinical visits was limited, especially during the COVID-19 pandemic.

At 16 months old, the patient presented to the clinic with features inconsistent with a diagnosis of mild hyperphenylalaninemia and not explained by prematurity alone, including significant hypotonia and developmental delay. She was beginning to sit up but not pulling to stand. Fine motor skills were normal but speech was delayed. No abnormal movements were noted. Dysmorphic features were also noted on examination, including brachycephaly and midface hypoplasia with a depressed nasal bridge. Exome sequencing was offered to exclude other genetic etiologies for these manifestations and specifically evaluated for causes of hyperphenylalaninemia, global developmental delay, hypotonia, and the listed dysmorphic features. Exome sequencing revealed that the patient was homozygous for a variant of *DNAJC12:* c. 235 C > T (p.R79 *). This variant was considered pathogenic by the testing laboratory due to it being a nonsense variant, predicted to result in a truncated protein product with loss of 120 downstream amino acids. Therefore, this genetic result was consistent with a diagnosis of *DNAJC12*-deficient hyperphenylalaninemia.

At 21 months of age, neurotransmitter deficiency relative to *DNAJC12*-deficient hyperphenylalaninemia was confirmed and showed severe reductions of 5-hydroxyindoleacetic acid (5-HIAA), homovanillic acid (HVA), and 3-O-Methyldopa (3-OMD). Prolactin levels were also measured, as prolactin levels are regulated by dopamine in a negative feedback loop [3]. Prolactin levels were elevated, indicating dopamine deficiency (see Table 1 for metabolite values measured during this assessment).

The management of *DNAJC12*-deficient hyperphenylalaninemia included maintaining phenylalanine levels within a therapeutic range and the supplementation of the deficient neurotransmitter precursors. Levodopa is the precursor to dopamine, epinephrine, and norepinephrine, and 5-hydroxytryptophan is a precursor to serotonin [4,5]. Therefore, therapy with sapropterin dihydrochloride, the synthetic form of the cofactor tetrahydrobiopterin (BH4); levodopa/carbidopa; and 5-hydroxytryptophan were initiated. The patient continued on an unrestricted diet and began physical and speech therapy at 23 months old. Since initiating treatment for *DNAJC12*-deficient hyperphenylalaninemia, the patient’s strength improved. The patient began walking with support (cruising) at 26 months with better ability to weight bear soon after initiating therapy, and she started walking independently at 3 years of age. Her speech is limited to only a few words. Additionally, she exhibits some behavioral differences, including poor attention and repetitive behaviors. However, the patient is interactive, and her presentation is not consistent with autism spectrum disorder, per her developmental pediatrician. There has been no regression of milestones. 

Exome sequencing also confirmed that the patient’s mother and father are both heterozygous for the *DNAJC12* c. 235 C > T pathogenic variant, thus making them both carriers of *DNAJC12*-deficient hyperphenylalaninemia. The patient’s mother and father share two additional children: the patient’s twin brother and an older sister. Both full-siblings screened negative on newborn screening for hyperphenylalaninemia, and confirmatory biochemical testing also revealed normal phenylalanine levels. Additionally, molecular testing was coordinated for her sister, which returned negative (see Figure 1 for the pedigree). 

## 3. Discussion

The protein encoded from the *DNAJC12* gene functions as a co-chaperone that participates in the proper folding of the biopterin-dependent aromatic amino acid hydroxylases, which include PAH, tyrosine hydroxylase, and peripheral and neuronal tryptophan hydroxylase [1,2]. The DNAJC12 protein was first identified in Drosophila and mice in 1999, and its human homolog was identified shortly afterwards in 2000, mapped on chromosome 10q21.1 [6,7]. It was initially discovered as part of a subgroup of proteins known as DNAJ proteins, which act as molecular chaperones of the HSP70 proteins (heat shock protein 70) [7]. The HSP70 proteins function as molecular chaperones that aid in the folding, remodeling, trafficking, and degradation of their target proteins, ensuring that the proper confirmation of these proteins is conserved for functionality [8,9]. DNAJ proteins are characterized by a highly conserved C-terminus and the J domain, which dictates the interactions of HSP70 proteins and specifies target proteins of interest [6]. The intrinsic ATPase activity in the HSP70 proteins is low, and they, therefore, rely on DNAJ proteins as co-chaperones to induce ATPase and stabilize interactions with target proteins [8]. In the case of *DNAJC12*, this function directs the HSP70 proteins to preserve the function of the biopterin-dependent AAAHs through the proper folding of these proteins [8,10]. Whether or not *DNAJC12* participates in the other functions of HSP70, including trafficking and degradation, has not been determined to the best of our knowledge. 

*DNAJC12*-deficient hyperphenylalaninemia was first reported in 2017 as a novel autosomal recessively inherited neurotransmitter disorder, characterized by deficiencies in dopamine, norepinephrine, and serotonin [1]. Since this time, over 40 patients have been reported with this condition in the literature [11]. At the time of its initial discovery, the characteristic phenotype of this deficiency included mild hyperphenylalaninemia, developmental delay, intellectual disability, hypotonia, and dystonia [2]. However, more patients have been reported since the initial report of this condition, and the associated phenotype has expanded. More severe features have now been associated with this condition, such as Parkinsonism, due to deficiencies in dopamine [12]. Behavioral concerns, such as autism spectrum disorder, attention deficit hyperactivity disorder, and psychiatric manifestations, have also been reported in some patients with *DNAJC12*-deficient hyperphenylalaninemia [10,12]. Conversely, less severe manifestations have been reported in some cases, including mild hypotonia or minor neurologic features as the only reported symptoms [12]. Additionally, some patients exhibit no symptoms outside of mild hyperphenylalaninemia [12]. However, data are limited on the asymptomatic and mild cases of *DNAJC12*-deficient hyperphenylalaninemia, as many of these patients did not have complete workups for the condition, including neuropsychiatric testing or CSF neurotransmitter analysis [12,13]. Due to the novelty of this condition, it is also undetermined whether asymptomatic patients may be at risk of later-onset manifestations, though it likely depends on the severity of neurotransmitter deficiency or phenylalanine elevations.

As per the initial reports of this condition, treatment for *DNAJC12*-deficient hyperphenylalaninemia includes the supplementation of deficient neurotransmitter precursors, restriction of phenylalanine to maintain blood phenylalanine levels within the therapeutic range, and administration of sapropterin dihydrochloride (BH4) [1,2]. As per the case report, these abnormalities can be determined by looking for elevations in phenylalanine in plasma and deficiencies in HVA, 5-HIAA, and 3-OMD in the CSF. However, with the expansion of the phenotype in recent history, the treatment regimens have also been expanded [13]. As is such, the current treatment regimens for patients with this condition include no treatments, supplementation with neurotransmitter precursors, supplementation with synthetic BH4, supplementation with both therapeutics, and supplementation with both therapeutics and phenylalanine restriction [10]. 

BH4 supplementation has been reported to normalize blood phenylalanine levels to a therapeutic range in all reported cases of *DNAJC12*-deficient hyperphenylalaninemia [14]. In these cases, phenylalanine restriction would be unnecessary if phenylalanine levels remain within therapeutic range with BH4 supplementation alone. BH4 supplementation has also been hypothesized to act both as a cofactor to facilitate the functioning of the AAAHs and as a chaperone to increase the stability of the enzymes [15]. Therefore, it has been proposed that BH4 acting as a chaperone in the absence of DNAJC12 ameliorates the deficit, and it may correct the deficit in the absence of neurotransmitter precursors [16]. However, the treatment of severe manifestations, such as Parkinsonism, with BH4 alone has not been explored. Given that Parkinsonism due to *DNAJC12*-deficiency is dopa-responsive, supplementation with levodopa/carbidopa is an established therapeutic to both treat and potentially prevent this manifestation [17]. Therefore, given the novelty of this condition and the spectrum of symptom severity, optimal treatment for patients with *DNAJC12*-deficient hyperphenylalaninemia remains to be defined [12].

Improvements in cognitive and motor functioning have been reported in patients with *DNAJC12*-deficient hyperphenylalaninemia after the initiation of therapies. Additionally, better outcomes have been reported when therapies are initiated earlier in life. A report involving three patients specified that initiating BH4 and levodopa/carbidopa with or without 5-hydroxytyptophan resulted in normal neurodevelopment when the treatment was initiated within the first year of life. In a separate study of three patients in which combined therapy was initiated after the second year of life, cognitive and motor functions improved, but normal neurodevelopment was not achieved [1,2,12,17]. Therefore, treatment initiated early in life is integral to both preventing deficits and maintaining normal development and cognitive function.

For our patient, though hyperphenylalaninemia was noted initially on newborn screening, the diagnosis of *DNAJC12*-deficient hyperphenylalaninemia was not made until she was 19 months old due to the prolonged NICU stay, limitations of molecular testing during the pandemic, and limited compliance with clinical visits during the pandemic. Notably, during the patient’s initial consultation, the most comprehensive multigene panel for hyperphenylalaninemia was ordered, which returned negative, as it did not include the *DNAJC12* gene. Since the time of the patient’s initial testing, the *DNAJC12* gene has been added to this panel; however, it is unclear if all babies with hyperphenylalaninemia undergo comprehensive molecular studies or even have access to this testing. 

Given this delay, the patient began sapropterin dihydrochloride supplementation at 21 months of age, levodopa/carbidopa supplementation at 23 months, and 5-hydroxytryptophan supplementation at 24 months of age. Prior to initiating therapies, her presentation was notable for significant hypotonia, motor delays, and speech delays. Fine motor skills were intact, and dystonic features were not noted on physical examination. Now, after taking therapeutics for over a year, the patient’s tone has improved, and she is catching up on her motor milestones. However, she is considered non-verbal and exhibits some behavioral differences, and her presentation is consistent with what has been described in the literature for patients with *DNAJC12*-deficient hyperphenylalaninemia who initiate therapies within the second year of life. However, the significant motor improvements should be highlighted, as the patient went from exhibiting difficulty ambulating to nearly normal motor milestones for her age. Therefore, it is evident that initiating therapies for *DNAJC12*-deficient hyperphenylalaninemia can result in significant improvement within this window of time.

## 4. Conclusions

*DNAJC12*-deficient hyperphenylalaninemia is a newly characterized inborn error of metabolism, and it should be considered on the differential for patients with hyperphenylalaninemia in the absence of *PAH* deficiency. This is especially important in cases of positive newborn screen results for hyperphenylalaninemia, as this condition presents with elevated phenylalanine in the absence of BH4 anomalies, consistent with *PAH* deficiency. *DNAJC12* molecular testing should be considered in all patients with hyperphenylalaninemia of unknown etiology, especially patients who screen positive on newborn screening, as the initiation of therapeutics early in life is integral to preventing manifestations and maintaining typical neurodevelopment. 

## Figures and Tables

**Figure 1 IJNS-10-00007-f001:**
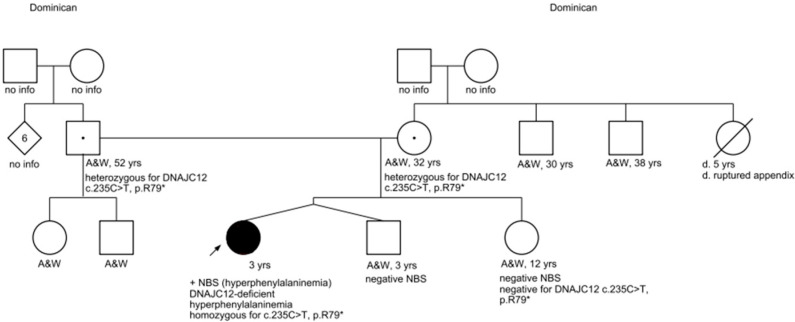
Pedigree. Key: A&W = alive and well.

**Table 1 IJNS-10-00007-t001:** CSF neurotransmitter levels and serum prolactin levels measured at 21 months of age, consistent with *DNAJC12*-deficient hyperphenylalaninemia.

Metabolite	Units	Reference	Patient’s Levels
5-HIAA	nmol/L	129–520	26 (Deficient)
HVA	nmol/L	294–1115	215 (Deficient)
3-OMD	nmol/L	<300	6
Prolactin	nmol/L	1.4–24	42.4 (Elevated)

## Data Availability

No new data were created or analyzed in this study. Data sharing is not applicable to this article.

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
