# Peer review of "A Case of DNAJC12-Deficient Hyperphenylalaninemia Detected on Newborn Screening: Clinical Outcomes from Early Detection"

_2409-515X, 2024, doi:10.3390/ijns10010007_

Round 1

Reviewer 1 Report

Comments and Suggestions for Authors

This manuscript is a case report of a patient with DNAJC12-related hyperphenylalaninemia who was missed following NBS due to failure to test for mutations in this gene when no variants were detected in PAH. Because of the rarity of this recently described disorder and the lack of familiarity of some clinicans, this will be a good reminder to the readership to consider this disorder in similar circumstances. There are some errors, typographical and others, in the mauscript that need to be corrected or clarified. These are listed in the section below. 

Comments on the Quality of English Language

p.2, line 59- the should be to

p2, line 63- the authors need to define what is meant by "cofactor screening"because I have no idea

p2, line 79- deficient is mispelled as deficienct

p3, line 96 - there is no subject in the sentence that begins with the word Began. 

p3, line 103- Should read There has (not have) been no regression of milestones 

p4, line 154- should read restriction of phenylalanine to reach blood phenylalanine levels within the therapeutic range

p4, line 182- is should be was

Author Response

Thank you for your suggestions. The spelling and grammatical checks were appreciated and have been addressed (comments 1,3,4,5, and 6). Instead of correcting "was" to "is" (p4, line 182, comment 7), the word "is" was corrected to "was" earlier in the paragraph (p4, line 181) for consistently in verb tense throughout the paragraph. Additional context to cofactor screening (comment 2) was included. Additional edits suggested from other reviewers include revisions to Table 1, further spelling/grammatical corrections, and adding a novel reference describing the number of patients with this condition per recent reports in the literature. 

Reviewer 2 Report

Comments and Suggestions for Authors

This case report of a child is very interesting, as hyperphenylalaninemia (HPA) is quite common, but the underlying DNAJC12-deficiency that appears to be present in some cases was first reported in 2017. So it is important to consider molecular testing in children with HPA and abnormal development, as early treatment might be important.

Some minor points:

Line 28-29: Please make sure to introduce abbreviations at the beginning and use only these afterwards. The abbreviations for these enzymes are only introduced in the discussion.

Line 31: In addition or additionally

Line 34-35: This sentence can be included in line 23-24: DNAJC12-deficient hyperphenylalaninemia is a newly described rare autosomal recessively inherited condition characterized by deficiencies in multiple…. Biallelic pathogenic variants….

Table 1: I would suggest to add the metabolic levels into the text, as this table does not provide much more information than the always elevated Phe-value and at only one timepoint other indicators.

Line 95-99: Very short sentences, maybe it would be better to summarize the information, i.e., line 95-96: The patient began cruising at 26 months with better ability to weight bear soon after initiating therapy and started walking independently at 3 years of age.

Figure 1: Please explain the abbreviations

Are both siblings tested negative (as described in line 105) or only the older sister (as shown in the figure)?

Line 120: Please explain the abbreviation HSP70

Discussion (line 132): How many patients have been described until today? Is it possible to estimate a prevalence?

Line 155: Is sapropterin dihydrochloride the same as sapropterin in line 95? And BH4 is used for the supplement and the cofactor?

Comments on the Quality of English Language

English language and style are fine/minor spell check required.

Author Response

Thank you for your suggestions. Please see below for our revisions:

Comment 1: Upon review, we removed abbreviations for the enzymes not consistently mentioned throughout the manuscript, including AAAHs, TH, TPH1, and TPH2. PAH continued to be used as an abbreviation and was defined in the "case report" subsection

Comments 2 and 3: Grammatical mistakes were addressed and these sentences were rearranged

Comment 4: The table is still included in this presentation to keep this information organized. However, this was adapted to only describe the neurotransmitter levels/prolactin levels and to include the reference ranges

Comment 5: Sentence structures revised as per suggestions

Comment 6: A key was added to Figure 1 to address abbreviations

Comment 7: further clarity to the extent of the family's testing (biochemical vs. molecular) was added to address this discrepancy

Comment 8: definition of HSP70 added

Comment 9: the current number of patients per the literature was added, including an additional reference (line 134-135)

Comment 10: edited line 95 to "sapropterin dihydrochloride" to provide consistency in language used throughout manuscript. "BH4 supplementation" is the language utilized by a number of other publications related to DNAJC12 deficiency. It is presumed that this is achieved through supplementation of sapropterin dihydrochloride; however, since this language was not explicitly used in other publications, the term "BH4 supplementation" was used in these paragraphs for consistency

Additional edits suggested from other reviewers include defining additional terms in the publication (e.g. cofactor screening, cruising) and correction of other spelling/grammatical errors. 

Reviewer 3 Report

Comments and Suggestions for Authors

A well written case report that adds to the knowledge base regarding newborn screening for hyperphenylalaninemias.  

I have only a few minor recommendations for the authors:

1.  Please add the names and addresses of the laboratoryies used for the initial multigene panel (l.57) and subsequent exome sequencing (l.75).

2. Please correct minor spelling errors - e.g. "cruising" should presumably read "crawling" (l.98)

3.  It would be appropriate to provide the total number of patients confirmed by DNA to have this condition according to the literature.

Author Response

Thank you for your suggestions. For comment 1, we elected to not include the identifying information of the commercial laboratories utilized during this patient's care to avoid bias (particularly, in which laboratories that we utilize more often compared to others). For comment 2, "cruising" refers to the patient walking with support or while holding objects, which was defined in the manuscript for further clarity. Finally, an additional reference was added to the manuscript to describe the number of patients currently identified with this condition, based on the literature (comment 3). Additional edits suggested from other reviewers include correcting spelling and grammatical errors, defining other terminology in the manuscript, and some revisions to Table 1.